# COMBINING PHYSICS AND MACHINE LEARNING FOR NETWORK FLOW ESTIMATION

**Arlei Silva, Furkan Kocayusufoglu**
Computer Science Department, UC Santa Barbara, CA 93106-5110, USA

**Saber Jafarpour, Francesco Bullo**
Mechanical Engineering Department and the Center of Control, Dynamical Systems and Computation, UC Santa Barbara, CA 93106-5070, USA

**Ananthram Swami**
U.S. Army Research Lab, Adelphi, MD 20783, USA

**Ambuj Singh**
Computer Science Department, UC Santa Barbara, CA 93106-5110, USA

## ABSTRACT

The flow estimation problem consists of predicting missing edge flows in a network (e.g., traffic, power, and water) based on partial observations. These missing flows depend both on the underlying *physics* (edge features and a flow conservation law) as well as the observed edge flows. This paper introduces an optimization framework for computing missing edge flows and solves the problem using bilevel optimization and deep learning. More specifically, we learn regularizers that depend on edge features (e.g., number of lanes in a road, resistance of a power line) using neural networks. Empirical results show that our method accurately predicts missing flows, outperforming the best baseline, and is able to capture relevant physical properties in traffic and power networks.

## 1 INTRODUCTION

In many applications, ranging from road traffic to supply chains to power networks, the dynamics of flows on edges of a graph is governed by physical laws/models (Bressan et al., 2014; Garavello & Piccoli, 2006). For instance, the LWR model describes equilibrium equations for road traffic Lighthill & Whitham (1955); Richards (1956). However, it is often difficult to fully observe flows in these applications and, as a result, they rely on off-the-shelf machine learning models to make predictions about missing flows (Li et al., 2017; Yu et al., 2018). A key limitation of these machine learning models is that they disregard the physics governing the flows. So, the question arises: can we combine physics and machine learning to make better flow predictions?

This paper investigates the problem of predicting missing edge flows based on partial observations and the underlying domain-specific physics defined by flow conservation and edge features (Jia et al., 2019). Edge flows depend on the graph topology due to a flow conservation law—i.e. the total in-flow at every vertex is approximately its total out-flow. Moreover, the flow at an edge also depends on its features, which might regularize the space of possible flow distributions in the graph. Here, we propose a model that learns how to predict missing flows from data using bilevel optimization (Franceschi et al., 2017) and neural networks. More specifically, features are given as inputs to a neural network that produces edge flow regularizers. Weights of the network are then optimized via reverse-mode differentiation based on a flow estimation loss from multiple train-validation pairs.

Our work falls under a broader effort towards incorporating physics knowledge to machine learning, which is relevant for natural sciences and engineering applications where data availability is limited (Rackauckas et al., 2020). Conservation laws (of energy, mass, momentum, charge, etc.) are

essential to our understanding of the physical world. The classical Noether's theorem shows that such laws arise from symmetries in nature (Hanc et al., 2004). However, flow estimation, which is an inverse problem (Tarantola, 2005; Arridge et al., 2019), is ill-posed under conservation alone. Regularization enables us to apply domain-knowledge in the solution of inverse problems.

We motivate our problem and evaluate its solutions using two application scenarios. The first is road traffic networks (Coclite et al., 2005), where vertices represent locations, edges are road segments, flows are counts of vehicles that traverse a segment and features include numbers of lanes and speed limits. The second scenario is electric power networks (Dörfler et al., 2018), where vertices represent power buses, edges are power lines, flows are amounts of power transmitted and edge features include resistances and lengths of lines. Irrigation channels, gas pipelines, blood circulation, supply chains, air traffic, and telecommunication networks are other examples of flow graphs.

Our contributions can be summarized as follows: (1) We introduce a missing flow estimation problem with applications in a broad class of flow graphs; (2) we propose a model for flow estimation that is able to learn the physics of flows by combining reverse-mode differentiation and neural networks; (3) we show that our model outperforms the best baseline by up to 18%; and (4) we provide evidence that our model learns interpretable physical properties, such as the role played by resistance in a power transmission network and by the number of lanes in a road traffic network.

## 2 FLOW ESTIMATION PROBLEM

We introduce the flow estimation problem, which consists of inferring missing flows in a network based on a flow conservation law and edge features. We provide a list of symbols in the Appendix.

**Flow Graph.** Let $\mathcal{G}(\mathcal{V}, \mathcal{E}, \mathcal{X})$ be a flow graph with vertices $\mathcal{V}$ ($n = |\mathcal{V}|$), edges $\mathcal{E}$ ($m = |\mathcal{E}|$), and edge feature matrix $\mathcal{X} \in \mathbb{R}^{m \times d}$, where $\mathcal{X}[e]$ are the features of edge $e$. A flow vector $\mathbf{f} \in \mathbb{R}^m$ contains the (possibly noisy) flow $f_e$ for each edge $e \in \mathcal{E}$. In case $\mathcal{G}$ is directed, $\mathbf{f} \in \mathbb{R}_+^m$, otherwise, a flow is negative if it goes against the arbitrary orientation of its edge. We assume that flows are induced by the graph, and thus, the total flow—in plus out—at each vertex is approximately conserved:

$$\sum_{(v_i, u) \in \mathcal{E}} f_{(v_i, u)} \approx \sum_{(u, v_o) \in \mathcal{E}} f_{(u, v_o)}, \forall u \in \mathcal{V}$$

In the case of a road network, flow conservation implies that vehicles mostly remain on the road.

**Flow Estimation Problem.** Given a graph $\mathcal{G}(\mathcal{V}, \mathcal{E}, \mathcal{X})$ with partial flow observations $\hat{\mathbf{f}} \in \mathbb{R}^{m'}$ for a subset $\mathcal{E}' \subseteq \mathcal{E}$ of edges ($\hat{f}_e$ is the flow for $e \in \mathcal{E}'$, $m' = |\mathcal{E}'| < m$), predict flows for edges in $\mathcal{E} \setminus \mathcal{E}'$.

In our road network example, partial vehicle counts $\hat{f}$ might be measured by sensors placed at a few segments, and the goal is to estimate counts at the remaining segments. One would expect flows not to be fully conserved in most applications due to the existence of inputs and outputs, such as parking lots and a power generators/consumers. In case these input and output values are known exactly, they can be easily incorporated to our problem as flow observations. Moreover, if they are known approximately, we can apply them as priors (as will be detailed in the next section). For the remaining of this paper, we assume that inputs and outputs are unknown and employ flow conservation as an approximation of the system. Thus, different from classical flow optimization problems, such as min-cost flow (Ahuja et al., 1988), we assume that flows are conserved approximately.

Notice that our problem is similar to the one studied in Jia et al. (2019). However, while their definition also assumes flow conservation, it does not take into account edge features. We claim that these features play important role in capturing the physics of flows. Our main contribution is a new model that is able to learn how to regularize flows based on edge features using neural networks.

## 3 OUR APPROACH: PHYSICS+LEARNING

In this section, we introduce our approach for the flow estimation problem, which is summarized in Figure 1. We formulate flow estimation as an optimization problem (Section 3.1), where the interplay between the flow network topology and edge features is defined by the physics of flow graphs. Flow estimation is shown to be equivalent to a regularized least-squares problem (Section

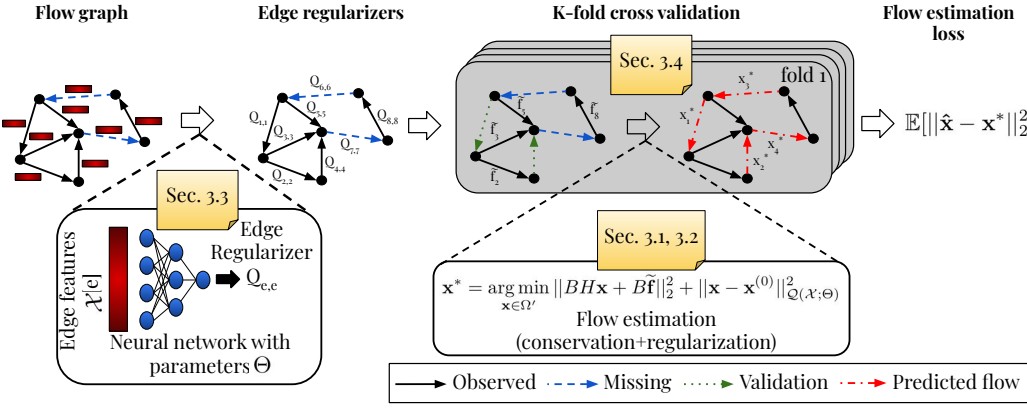

Figure 1: Summary of the proposed approach for predicting missing flows in a graph based on partial observations and edge features. We learn to combine features and a flow conservation law, which together define the physics of the flow graph. A regularization function $\mathcal{Q}(\mathcal{X};\Theta)$ modeled as a neural network with parameters $\Theta$ takes as input edge features $\mathcal{X}[e]$. A flow estimation algorithm applies the regularization, partial observations ($\widetilde{\mathbf{f}}$), prior flows ($\mathbf{x}^{(0)}$) and flow conservation to predict missing flows $\mathbf{x}$. Network parameters $\Theta$ are learned based on a $K$-fold cross validation loss with respect to validation flows $\hat{\mathbf{x}}$. Our model is trained end-to-end using reverse-mode differentiation.

3.2). Moreover, we describe how the effect of edge features and the graph topology can be learned from data using bilevel optimization and neural networks in Section 3.3. Finally, we propose a reverse-mode differentiation algorithm for flow estimation in Section 3.4.

## 3.1 FLOW ESTIMATION VIA OPTIMIZATION

The extent to which flow conservation holds for flows in a graph is known as *divergence* and can be measured using the *oriented incidence matrix* $B \in \mathbb{R}^{n \times m}$ of $\mathcal{G}$. The matrix is defined as follows, $B_{ij} = 1$ if $\exists u$ such that $e_j = (v_i, u) \in \mathcal{E}$, $B_{ij} = -1$ if $\exists u$ such that $e_j = (u, v_i) \in \mathcal{E}$, and $B_{ij} = 0$, otherwise. Given $B$ and $\mathbf{f}$, the divergence at a vertex $u$ can be computed as:

$$(B\mathbf{f})_u = \sum_{(v_i, u) \in \mathcal{E}} f_{(v_i, u)} - \sum_{(u, v_o) \in \mathcal{E}} f_{(u, v_o)} \tag{1}$$

And thus, we can compute the total (squared) divergence in the graph as $||B\mathbf{f}||_2^2 = \mathbf{f}^\intercal B^\intercal B \mathbf{f} = \sum_{u \in \mathcal{V}} ((B\mathbf{f})_u)^2$. One could try to solve the flow estimation problem by minimizing $||B\mathbf{f}||_2^2$ while keeping the observed flows fixed, however, this problem is ill-posed—there might be multiple solutions to the optimization. The standard approach in such a scenario is to resort to regularization. In particular, we apply a generic regularization function $\Phi$ with parameters $\Theta$ as follows:

$$\mathbf{f}^* = \underset{\mathbf{f} \in \Omega}{\arg\min} ||B\mathbf{f}||_2^2 + \Phi(\mathbf{f}, \mathcal{X}; \mathbf{f}^{(0)}; \Theta) \quad st. \quad f_e = \hat{f}_e, \forall e \in \mathcal{E}' \tag{2}$$

where $\Omega$ is the domain of $\mathbf{f}$, $\mathbf{f}^{(0)} \in \mathbb{R}^m$ is a prior for flows, $f_e$ ($\hat{f}_e$) are entries of $\mathbf{f}$ ($\hat{\mathbf{f}}$) for edge $e$ and the constraint guarantees that observed flows are not changed. Priors $\mathbf{f}^{(0)}$, not be confused with observed flows $\hat{\mathbf{f}}$, should be set according to the application (e.g., as zero, based on a black-box model or historical data). Regarding the domain $\Omega$, we consider $\Omega = \mathbb{R}^m$ and $\Omega = \mathbb{R}^m_+$. The second case is relevant for directed graphs—when flows must follow edge orientations (e.g., traffic).

In Jia et al. (2019), the authors set $\Phi(\mathbf{f}, X, \mathbf{f}^{(0)}; \Theta)$ as $\lambda^2 ||\mathbf{f}||_2^2$ for a regularization parameter $\lambda$, which implies a uniform zero prior with an $L_2$ penalty over edges. We claim that the regularization function plays an important role in capturing the physics of flow graphs. As an example, for a power network, $\Phi$ should account for the resistance of the lines. Thus, we propose learning the regularization from data. Our approach is based on a least-squares formulation, which will be described next.

### 3.2 REGULARIZED LEAST-SQUARES FORMULATION

Flow estimation problem can be viewed as an *inverse problem* (Tarantola, 2005). Let $\mathbf{x} \in \mathbb{R}^{m-m'}$ be the vector of missing flows and $H \in \mathbb{R}^{m \times m-m'}$ be a matrix such that $H_{ij} = 1$ if $f_i$ maps to $x_j$ (i.e., they are associated to the same edge), and $H_{i,j} = 0$, otherwise. Moreover, let $\widetilde{\mathbf{f}} \in \mathbb{R}^m$ be such that $\widetilde{f}_e = \hat{f}_e$ if $e \in \mathcal{E}'$ and $\widetilde{f}_i = 0$, otherwise. Using this notation, we define flow estimation as $BH\mathbf{x} = -B\widetilde{\mathbf{f}} + \epsilon$, where $BH$ is a forward operator, projecting $\mathbf{x}$ to a vector of vertex divergences, and $-B\widetilde{\mathbf{f}} + \epsilon$ is the observed data, capturing (negative) vertex divergences for observed flows. The error $\epsilon$ can be interpreted as noise in observations or some level of model misspecification.

We can also define a regularized least-squares problem with the goal of recovering missing flows $\mathbf{x}$:

$$\mathbf{x}^* = \underset{\mathbf{x} \in \Omega'}{\arg\min} ||BH\mathbf{x} + B\widetilde{\mathbf{f}}||_2^2 + ||\mathbf{x} - \mathbf{x}^{(0)}||_{\mathcal{Q}(\mathcal{X};\Theta)}^2 \tag{3}$$

where $\Omega'$ is a projection of the domain of $\mathbf{f}$ to the space of $\mathbf{x}$, $||\mathbf{x}||_M^2 = \mathbf{x}^\intercal M\mathbf{x}$ is the matrix-scaled norm of $\mathbf{x}$ and $\mathbf{x}^{(0)} \in \mathbb{R}^{m-m'}$ are priors for missing flows. The regularization function $\Phi(\mathbf{f}, \mathcal{X}; \mathbf{f}^{(0)}, \Theta)$ has the form $||\mathbf{x} - \mathbf{x}^{(0)}||_{\mathcal{Q}(\mathcal{X};\Theta)}^2$, where the matrix $\mathcal{Q}(\mathcal{X};\Theta)$ is a function of parameters $\Theta$ and edge features $\mathcal{X}$. We focus on the case where $\mathcal{Q}(\mathcal{X};\Theta)$ is non-negative and diagonal.

Equation 3 has a Bayesian interpretation, with $\mathbf{x}$ being a maximum likelihood estimate under a Gaussian assumption—i.e., $\mathbf{x} \sim N(\mathbf{x}^{(0)}, \mathcal{Q}(\mathcal{X};\Theta)^{-1})$ and $B\widetilde{\mathbf{f}} \sim N(0, I)$ (Tarantola, 2005). Thus, $\mathcal{Q}(\mathcal{X};\Theta)$ captures the variance in flow observations $\hat{\mathbf{f}}$ in prior estimates $\mathbf{f}^{(0)}$ compared to the one. This allows the regularization function to adapt to different edges based on their features. For instance, in our road network example, $Q(\mathcal{X};\Theta)$ might place a lower weight on flow conservation for flows at a road segment with a small number of lanes, which are possible traffic bottlenecks.

Given the least-squares formulation described in this section, how do we model the regularization function $\mathcal{Q}$ and learn its parameters $\Theta$? We would like $\mathcal{Q}$ to be expressive enough to be able to capture complex physical properties of flows, while $\Theta$ to be computed accurately and efficiently. We will address these challenges in the remaining of this paper.

### 3.3 BILEVEL OPTIMIZATION FOR META-LEARNING THE PHYSICS OF FLOWS

This section introduces a model for flow estimation that is able to learn the regularization function $\mathcal{Q}(\mathcal{X};\Theta)$ in Equation 3 from data using bilevel optimization and neural networks.

**Bilevel formulation.** We learn the parameters $\Theta$ that determine the regularization function $\mathcal{Q}(\mathcal{X};\Theta)$ using the following bilevel optimization formulation:

$$\Theta^* = \underset{\Theta}{\arg\min} \mathbb{E}[||\hat{\mathbf{x}} - \mathbf{x}^*||_2^2] \tag{4}$$

$$\text{st.} \quad \mathbf{x}^* = \underset{\mathbf{x} \in \Omega'}{\arg\min} ||BH\mathbf{x} + B\widetilde{\mathbf{f}}||_2^2 + ||\mathbf{x} - \mathbf{x}_0||_{Q(\mathcal{X};\Theta)}^2 \tag{5}$$

where the inner (lower) problem is the same as Equation 3 and the outer (upper) problem is the expected loss with respect to ground truth flows $\hat{\mathbf{x}}$—which we estimate using cross-validation.

Notice that optimal values for parameters $\Theta$ and missing flows $\mathbf{x}$ are both unknown in the bilevel optimization problem. The expectation in Equation 4 is a function of multiple instances of the inner problem (Equation 5). Each inner problem instance has an optimal solution $\mathbf{x}^*$ that depends on parameters $\Theta$. In general, bilevel optimization is not only non-convex but also NP-hard (Colson et al., 2007). However, recent gradient-based solutions for bilevel optimization have been successfully applied to large-scale problems, such as hyper-parameter optimization and meta-learning (Franceschi et al., 2018; Lorraine et al., 2020). We will first describe how we model the function $\mathcal{Q}(\mathcal{X};\Theta)$ and then discuss how this problem can be solved efficiently using reverse-mode differentiation.

We propose to model $\mathcal{Q}(\mathcal{X};\Theta)$ using a neural network, where $\mathcal{X}$ are inputs, $\Theta$ are learnable weights and the outputs are diagonal entries of the regularization matrix. This is a natural choice due to the expressive power of neural nets (Cybenko, 1989; Xu et al., 2018).

**Multi-Layer Perceptron (MLP).** An MLP-based $\mathcal{Q}(\mathcal{X};\Theta)$ has the following form:

$$\mathcal{Q}(\mathcal{X};\Theta) = diag(MLP(\mathcal{X};\Theta)) \tag{6}$$

where $MLP(\mathcal{X}; \Theta) \in \mathbb{R}^{m-m'}$. For instance, $\mathcal{Q}(\mathcal{X}; \Theta)$ can be a 2-layer MLP:

$$\mathcal{Q}(\mathcal{X}; \Theta) = diag(a(b(\mathcal{X}W^{(1)})W^{(2)})) \qquad (7)$$

where $\Theta = \{W^{(1)}, W^{(2)}\}$, $W^{(1)} \in \mathbb{R}^{d \times h}$, $W^{(2)} \in \mathbb{R}^{h \times 1}$, $h$ is the number of nodes in the hidden layer, both $a$ and $b$ are activation functions, and the bias was omitted for convenience.

**Graph Neural Network (GNN).** The MLP-based approach assumes that each entry $[\mathcal{Q}(\mathcal{X}; \Theta)]_{e,e}$ associated to an edge $e$ is a function of its features $\mathcal{X}[e]$ only. However, we are also interested in how entries $[\mathcal{Q}(\mathcal{X}; \Theta)]_{e,e}$ might depend on the features of neighborhood of $e$ in the flow graph topology. Thus, we consider the case where $\mathcal{Q}(\mathcal{X}; \Theta)$ is a GNN, which is described in the Appendix.

### 3.4    FLOW ESTIMATION ALGORITHM

We now focus on how to solve our bilevel optimization problem (Equations 4 and 5). Our solution applies gradient-based approaches (e.g., SGD (Bottou & Bousquet, 2008), Adam (Kingma & Ba, 2014)) and, for simplicity, our description will be based on the particular case of Gradient Descent and assume a zero prior ($\mathbf{x}^{(0)} = \mathbf{0}$). A key challenge in our problem is to efficiently approximate the gradient of the outer objective with respect to the parameters $\Theta$, which, by the chain rule, depends on the gradient of the inner objective with respect to $\Theta$.

We first introduce extra notation to describe the outer problem (Equation 4). Let $(\hat{\mathbf{f}}_k, \hat{\mathbf{g}}_k)$ be one of $K$ train-validation folds, both containing ground-truth flow values, such that $\hat{\mathbf{f}}_k \in \mathbb{R}^p$ and $\hat{\mathbf{g}}_k \in \mathbb{R}^q$. For each fold $k$, we apply the inner problem (Equation 5) to estimate missing flows $\mathbf{x}_k$. Estimates for all folds are concatenated into a single vector $\mathbf{x} = [\mathbf{x}_1; \mathbf{x}_2; \dots; \mathbf{x}_K]$ and the same for validation sets $\hat{\mathbf{g}} = [\hat{\mathbf{g}}_1; \hat{\mathbf{g}}_2; \dots \hat{\mathbf{g}}_K]$. We define a matrix $R \in \mathbb{R}^{q \times (m-m')}$ such that $R_{ij} = 1$ if prediction $x_j$ corresponds to validation flow $\hat{g}_i$ and $R_{ij} = 0$, otherwise. Using this representation, we can approximate the expectation in the outer objective as $\Psi(\mathbf{x}, \Theta) = (1/K)||R\mathbf{x} - \hat{\mathbf{g}}||_2^2$, where $\mathbf{x}$ depends implicitly on $\Theta$. We also introduce $\Upsilon_\Theta(\mathbf{x})$ as the inner problem objective. Moreover, let $\Gamma_j(\mathbf{x}_{k,j-1}, \Theta_{i-1})$ be one step of gradient descent for the value of $\mathbf{x}_k$ at iteration $j$ with learning rate $\beta$:

$$\begin{aligned}
\Gamma_j(\mathbf{x}_{k,j-1}, \Theta_{i-1}) &= \mathbf{x}_{k,j-1} - \beta \nabla_\mathbf{x} \Upsilon_\Theta(\mathbf{x}_{k,j}) \\
&= \mathbf{x}_{k,j-1} - 2\beta[H_k^\mathsf{T} B^\mathsf{T}(BH_k\mathbf{x}_{k,j-1} + B\widetilde{\mathbf{f}}_k) + 2Q_k\mathbf{x}_{k,j-1}]
\end{aligned}$$

where $H_k$, $Q_k$ and $\widetilde{\mathbf{f}}_k$ are the matrix $H$, a sub-matrix of $\mathcal{Q}(\mathcal{X}; \Theta_{i-1})$ and the observed flows vector $\widetilde{\mathbf{f}}$ (see Section 3.2) for the specific fold $k$. We have assumed the domain ($\Omega'$) of flows $\mathbf{x}_{k,j}$ to be the set of real vectors. For non-negative flows, we add the appropriate proximal operator to $\Gamma_j$.

Our algorithm applies *Reverse-Mode Differentiation* (RMD) (Domke, 2012; Franceschi et al., 2017) to estimate $\nabla_\Theta \Psi$ and optimizes $\Theta$ also using an iterative algorithm. The main idea of RMD is to first unroll and store a finite number of iterations for the inner problem $\mathbf{x}_1, \mathbf{x}_2, \dots \mathbf{x}_J$ and then reverse over those iterations to estimate $\nabla_\Theta \Psi$, which is computed as follows:

$$\nabla_{\mathbf{x}_J, \Theta} \Psi(\mathbf{x}_J, \Theta_i) = \nabla_\mathbf{x} \Psi(\mathbf{x}_J, \Theta_i) \sum_{j=1}^{J} \left( \prod_{s=j+1}^{J} \frac{\partial \Gamma_s(\mathbf{x}_{s-1}, \Theta_i)}{\partial \mathbf{x}_{s-1}} \right) \frac{\partial \Gamma_j(\mathbf{x}_{j-1}, \Theta_i)}{\partial \Theta}$$

In particular, our reverse iteration is based on the following equations:

$$\nabla_\mathbf{x} \Psi(\mathbf{x}_J, \Theta_i) = (2/K)R^\mathsf{T}(R\mathbf{x}_J - \hat{\mathbf{g}})$$

$$\frac{\partial \Gamma_s(\mathbf{x}_{s-1}, \Theta_i)}{\partial \mathbf{x}_{s-1}} = I - 2\beta(H^\mathsf{T} B^\mathsf{T} BH + 2\mathcal{Q}(\mathcal{X}; \Theta_i))$$

$$\frac{\partial \Gamma_j(\mathbf{x}_{j-1}, \Theta_i)}{\partial \Theta} = -4\beta(\partial \mathcal{Q}(\mathcal{X}; \Theta_i)/\partial \Theta)\mathbf{x}_{j-1}$$

where $\partial \mathcal{Q}(\mathcal{X}; \Theta_i)/\partial \Theta$ is the gradient of the regularization function $\mathcal{Q}(\mathcal{X}; \Theta)$ evaluated at $\Theta_i$. In our case, this gradient is the same as the neural network gradients and is omitted here for convenience.

Algorithm 1 describes our RMD approach for flow estimation. It receives as inputs the flow network $\mathcal{G}(\mathcal{V}, \mathcal{E}, \mathcal{X})$, $K$ train-validation folds $\{(\hat{\mathbf{f}}_k, \hat{\mathbf{g}}_k)\}_{k=1}^{K}$, and also hyperparameters $T$, $J$, $\alpha$, and $\beta$,

---

**Algorithm 1** RMD Algorithm for Flow Estimation

---

**Require:** Flow network $\mathcal{G}(\mathcal{V}, \mathcal{E}, \mathcal{X})$, train-validation folds $\{(\hat{\mathbf{f}}_k, \hat{\mathbf{g}}_k)\}_{k=1}^K$, number of outer iterations $T$ and inner iterations $J$, learning rates $\alpha$ and $\beta$
**Ensure:** Regularization parameters $\Theta$
 1: Initialize parameters $\Theta_0$
 2: $\hat{\mathbf{g}} \leftarrow [\hat{\mathbf{g}_1}; \ldots \hat{\mathbf{g}_K}]$
 3: $B \leftarrow$ incidence matrix of $\mathcal{G}$
 4: **for** outer iterations $i = 1, \ldots T$ **do**
 5:     Initialize missing flows $\mathbf{x}_{k,0}$ for all $k$
 6:     **for** inner iterations $j = 1, \ldots J$ **do**
 7:         **for** folds $k = 1, \ldots K$ **do**
 8:             $\mathbf{x}_{k,j} \leftarrow \mathbf{x}_{k,j-1} - 2\beta[H_k^\intercal B^\intercal (BH_k\mathbf{x}_{k,j-1} + B\widetilde{\mathbf{f}}_k) + 2Q_k\mathbf{x}_{k,j-1}]$
 9:         **end for**
10:         $\mathbf{x}_j \leftarrow [\mathbf{x}_{1,j}; \ldots \mathbf{x}_{K,j}]$
11:     **end for**
12:     $\mathbf{z}_J \leftarrow (2/K)R^T(R\mathbf{x}_J - \hat{\mathbf{g}})$
13:     **for** reverse inner iterations $j = J-1, \ldots 1$ **do**
14:         $\overleftarrow{\Theta} \leftarrow \overleftarrow{\Theta} - 4\beta\mathbf{z}_{j+1}(\partial\mathcal{Q}(\mathcal{X}; \Theta_{i-1})/\partial\Theta)\mathbf{x}_{j+1}$
15:         $\mathbf{z}_j \leftarrow \mathbf{z}_{j+1}[I - 2\beta(H^\intercal B^\intercal BH + \mathcal{Q}(\mathcal{X}; \Theta_{i-1}))]$
16:     **end for**
17:     Update $\Theta_i \leftarrow \Theta_{i-1} - \alpha\overleftarrow{\Theta}$
18: **end for**
19: **return** parameters $\Theta_I$

---

corresponding to the number of outer and inner iterations, and learning rates for the outer and inner problem, respectively. Its output is a vector of optimal parameters $\Theta$ for the regularization function $\mathcal{Q}(\mathcal{X}; \Theta)$ according to the bilevel objective in Equations 4 and 5. We use $\overleftarrow{\Theta}$ to indicate our estimate of $\nabla_\Theta\Psi(\Theta_i)$. Iterations of the inner problem are stored for each train-validation fold in lines 4-12. Reverse steps, which produce an estimate $\overleftarrow{\Theta}$, are performed in lines 13-17. We then use $\overleftarrow{\Theta}$ to update our estimate of $\Theta$ in line 17. The time and space complexities of the algorithm are $O(TJKm)$ and $O(Jm)$, respectively, due to the cost of computing and storing the inner problem iterations.

As discussed in the previous section, bilevel optimization is non-convex and thus we cannot guarantee that Algorithm 1 will return a global optima. In particular, the learning objective of our regularization function $\mathcal{Q}(\mathcal{X}; \Theta)$ is non-convex—it is a neural network. However, the inner problem (Equation 5) in our formulation has a convex objective (least-squares). In Franceschi et al. (2018), the authors have shown that this property implies convergence. We also find that our algorithm often converges to a good estimate of the parameters in our experiments.

## 4 EXPERIMENTS

We evaluate our approaches for the flow estimation problem using two real datasets and a representative set of baselines and metrics. Due to space limitations, we provide an extended version of this section, with more details on datasets, experimental settings, and additional results in the Appendix.

### 4.1 DATASETS

This section summarizes the datasets used in our evaluation. We normalize flow values to $[0, 1]$ and map discrete features to real vector dimensions using one-hot encoding.

**Traffic:** Vertices represent locations and directed edges represent road segments between two locations in Los Angeles County, CA. Flows are daily average vehicle counts measured by sensors placed along highways in the year 2018. We assign each sensor to an edge in the graph based on proximity and other sensor attributes. Our road network covers the Los Angeles County area, with $5,749$ vertices, $7,498$ edges, of which $2,879$ edges ($38\%$) have sensors. The following features were mapped to an 18-dimensional vector: lat-long coordinates, number of lanes, max-speed, and

highway type (motorway, motorway link, trunk, etc.), in-degree, out-degree, and centrality (PageRank). The in-degree and centrality of an edge are computed based on its source vertex. Similarly, the out-degree of an edge is the out-degree of its target vertex.

**Power:** Vertices represent buses in Europe, undirected edges are power transmission lines and edge flows measure the total active power (in MW) being transmitted through the lines. The dataset is obtained from PyPSA-Eur (Hörsch et al., 2018; Brown et al., 2017)—an optimization model of the European power transmission system—which generates realistic power flows based on solutions of optimal linear power flow problems with historical production and consumption data. Default values were applied for the PyPSA-Eur settings. The resulting graph has 2,048 vertices, 2,729 edges, and 14-dimensional feature vectors capturing resistance, reactance, length, and number of parallel lines, nominal power, edge degree etc. Please see the Appendix for more details.

## 4.2 EXPERIMENTAL SETTINGS

**Evaluation metrics:** We apply Pearson's correlation (CORR), Mean Absolute Percentage Error (MAPE), Mean Absolute Error (MAE), and Root Mean Squared Error (RMSE) to compare ground-truth and predicted flows. These metrics are formally defined in the Appendix.

**Baselines:** Divergence minimization (*Div*) (Jia et al., 2019) maximizes flow conservation using a single regularization parameter $\lambda$, which we optimize using line search in a validation set of flows. Multi-Layer Perceptron (*MLP*) is a 2-layer neural network with ReLU activations for all layers that learns to predict flows based on edge features. Graph Convolutional Network (*GCN*) is a 2-layer graph neural network, also with ReLU activations and Chebyshev convolutions of degree 2, that learns to predict the flows using both edge features and the topology but disregarding flow conservation (Kipf & Welling, 2016; Defferrard et al., 2016). We also consider two hybrid baselines. *MLP-Div* applies the predictions from *MLP* as priors to *Div*. Similarly, predictions from *GCN* are used as priors for *GCN-Div*. For both hybrid models, we also optimize the parameter $\lambda$.

**Our approaches:** We consider three variations of Algorithm 1. However, one important modification is that we perform the reverse iterations for each fold—i.e., folds are treated as batches in SGD. Bil-MLP and Bil-GCN apply our reverse-mode differentiation approach using an MLP and a GCN as a regularizer. Moreover, both approaches use zero as the prior $\mathbf{x}^{(0)}$. Bil-GCN-Prior applies the GCN predictions as flow priors. Architectures of the neural nets are the same as the baselines.

## 4.3 FLOW ESTIMATION ACCURACY

Table 1 compares our methods and the baselines in terms of several metrics using the Traffic and Power datasets. Values of CORR achieved by MLP and GCN for Traffic are missing because they were undefined—they have generated predictions with zero variance for at least one of the train-test folds. All methods suffer from high MAPE errors for Power, which is due to an over-estimation of small flows. Bil-GCN achieves the best results in both datasets in terms of all metrics, with 6% and 18% lower RMSE than the best baseline for Traffic and Power, respectively. However, notice that Bil-MLP and Bil-GCN achieve very similar performance for Power and Bil-GCN-Prior does not outperform our other methods. We also show scatter plots with the true vs. predicted flows for some of the best approaches in Figure 2. Traffic has shown to be the more challenging dataset, which can be explained, in part, by training data sparsity—only 38% of edges are labeled.

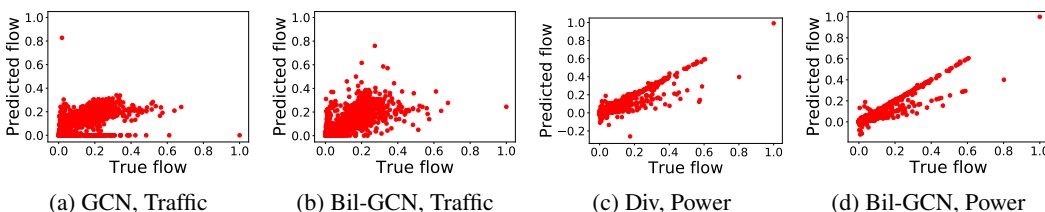

|  (a) GCN, Traffic | (b) Bil-GCN, Traffic | (c) Div, Power | (d) Bil-GCN, Power |

Figure 2: Scatter plots with true (x) and predicted (y) flows for two approaches on each dataset. The results are consistent with Table 1 and show that our methods are more accurate than the baselines.

| Method | Traffic | | | | Power | | | |
|---|---|---|---|---|---|---|---|---|
| | RMSE | MAE | MAPE | CORR | RMSE | MAE | MAPE | CORR |
| Div | 0.071 | 0.041 | 1.23 | 0.76 | 0.034 | 0.015 | 1419.2 | 0.93 |
| MLP | 0.083 | 0.055 | 1.13 | - | 0.069 | 0.043 | 8334.5 | 0.61 |
| GCN | 0.066 | 0.040 | 0.94 | - | 0.064 | 0.043 | 5622.3 | 0.64 |
| MLP-Div | 0.066 | 0.041 | 1.51 | 0.81 | 0.033 | 0.015 | 1593.5 | 0.93 |
| GCN-Div | 0.071 | 0.048 | 1.69 | 0.81 | 0.033 | 0.015 | 1795.2 | 0.93 |
| Bil-MLP | 0.069 | 0.038 | 1.05 | 0.79 | 0.027 | 0.011 | 758.0 | 0.95 |
| Bil-GCN | 0.062 | 0.034 | 0.86 | 0.82 | 0.027 | 0.011 | 788.5 | 0.95 |
| Bil-GCN-Prior | 0.062 | 0.035 | 0.91 | 0.82 | 0.027 | 0.011 | 691.5 | 0.95 |

Table 1: Average flow estimation accuracy for the baselines (Div, MLP and GCN) and our methods (Bil-MLP, Bil-GCN and Bil-GCN-Prior) using the Traffic and Power datasets. RMSE, MAE and MAPE are errors (the lower the better) and CORR is a correlation (the higher the better). Values of correlation for MLP and GCN using Traffic were undefined. Bil-GCN (ours) outperforms the best baseline for all the metrics, with up to 20% lower RMSE than Div using Power.

## 4.4 ANALYSIS OF REGULARIZERS

Figure 3 illustrates the regularization function learned by Bil-MLP. We focus on Bil-MLP because it can be analyzed independently of the topology. Figures 3a-3c show scatter plots where the x and y axes represent the value of the regularizer and features, respectively. For Power, Bil-MLP captures the effect of resistance over flows (Fig. 3a). However, only high values of resistance are mostly affected—that is the reason few points can be seen and also explains the good results for Div. We did not find a significant correlation for other features, with the exception of reactance, which is related to resistance. For Traffic, the model learns how the number of lanes constrains the flow at a road segment (Fig. 3b). Results for speed limit are more surprising, 45mph roads are less regularized (Fig. 3c). This is evidence that regularization is affecting mostly traffic bottlenecks in highways—with few lanes but a 65mph speed limit. To further investigate this result, we also show the regularizers over the Traffic topology in Figure 3d. High regularization overlaps with well-known congested areas in Los Angeles, CA (e.g., Highway 5, Southeast). These results are strong evidence that our methods are able to learn the physics of flows in road traffic and power networks.

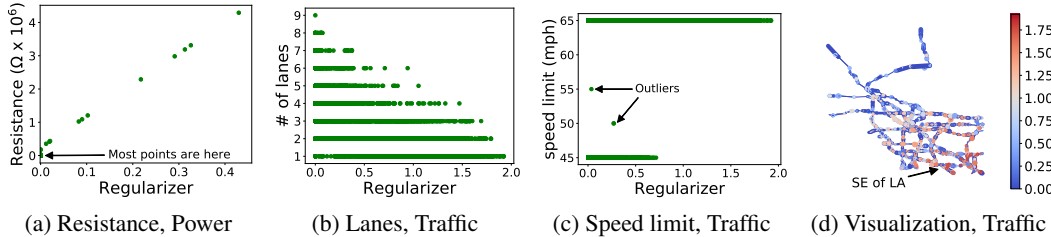

(a) Resistance, Power     (b) Lanes, Traffic     (c) Speed limit, Traffic     (d) Visualization, Traffic

Figure 3: Edge regularizer learned by Bil-MLP vs. features values (a-c) and visualization of regularizers on the Traffic topology (d). Our model is able to learn the effect of the resistance for Power. In Traffic, a higher number of lanes is correlated to less regularization and lower speed roads (45mph) are less regularized. The regularization is also correlated with congested areas in Los Angeles, CA.

## 5 RELATED WORK

Flow graphs are quite ubiquitous in engineering, biomedical and social sciences. Two important properties of flow graphs are that their state space is defined by a graph topology and their dynamics are governed by the physics (or logic) of the problem of interest. We refer to Bressan et al. (2014) for a unified characterization of the mathematical treatment of flow graphs. Notice that these studies do not address the flow inference problem and their applications to real data is limited (Herrera et al.,

2010; Work et al., 2010). Moreover, we focus on long term flows (e.g. daily vehicle traffic flows) and not on the dynamics. This simplifies the equations of our model to the conservation law.

Flow inference via divergence minimization was originally proposed in Jia et al. (2019). However, their work has not considered edge features and instead applied a single regularization parameter to the norm of the flow vector $\mathbf{f}$ in Equation 2. Our work leverages relevant edge features to learn the interplay between flow conservation and local predictions (priors). Thus, we generalize the formulation from Jia et al. (2019) to the case of a learnable regularization function $\mathcal{Q}(\Theta, X)$. Our experiments show that the proposed approach achieves superior results in two datasets.

Flow optimization problems, such as min-cost flow, max-flow and multi-commodity flow, have a long history in computer science (Ahuja et al., 1988; Ford Jr & Fulkerson, 2015). These problems impose flow conservation as a hard constraint, requiring full knowledge of source and sink vertices and noiseless flow observations. Our approach relaxes these requirements by minimizing the flow divergence (see Equation 2). Moreover, our problem does not assume edge capacities and costs.

The relationship between flow estimation and inverse problems is of particular interest due to the role played by regularization (Engl et al., 1996) in the solution of ill-posed problems. Recent work on inverse problems has also focused on learning to regularize based on data and even learning the forward operator as well—see Arridge et al. (2019) for a review. The use of the expression "learning the physics" is also popular in the context of the universal differential equation framework, which enables the incorporation of domain-knowledge from scientific models to machine learning (Raissi et al., 2019; Long et al., 2018; Rackauckas et al., 2020).

Bilevel optimization in machine learning has been popularized due its applications in hyperparameter optimization (Bengio, 2000; Larsen et al., 1996). In the last decade, deep learning has motivated novel approaches able to optimize millions of hyperparameters using gradient-based schemes (Maclaurin et al., 2015; Lorraine et al., 2020; Pedregosa, 2016). Our flow estimation algorithm is based on reverse-mode differentiation, which is a scalable approach for bilevel optimization (Franceschi et al., 2017; Domke, 2012; Maclaurin et al., 2015). Another application of bilevel optimization quite related to ours is meta-learning (Franceschi et al., 2018; Grefenstette et al., 2019).

Our problem is also related to semi-supervised learning on graphs (Zhu et al., 2003; Belkin et al., 2006; Zhou et al., 2004), which is the inference of vertex labels given partial observations. These approaches can be applied for flow estimation via the line graph transformation (Jia et al., 2019). The duality between a recent approach for predicting vertex labels Hallac et al. (2015) and min-cost flows was shown in Jung (2020). However, the same relation does not hold for flow estimation.

Graph neural network models, which generalize deep learning to graph data, have been shown to outperform traditional semi-supervised learning methods in many tasks (Kipf & Welling, 2016; Hamilton et al., 2017; Veličković et al., 2018). These models have also been applied for traffic forecasting (Li et al., 2017; Yu et al., 2018; Yao et al., 2019). Different from our approach, traditional GNNs do not conserve flows. We show that our models outperform GNNs at flow prediction. Moreover, we also apply GNNs as a regularization function in our model.

## 6 CONCLUSIONS

We have introduced an approach for flow estimation on graphs by combining a conservation law and edge features. Our model learns the physics of flows from data by combining bilevel optimization and deep learning. Experiments using traffic and power networks have shown that the proposed model outperforms a set of baselines and learns interpretable physical properties of flow graphs.

While we have focused on learning a diagonal regularization matrix, we want to apply our framework to the case of a full matri. We are also interested in combining different edge measurements in order to learn more complex physical laws, such as described by the fundamental diagram in the LWR model Lighthill & Whitham (1955); Daganzo (1994; 1995); Garavello & Piccoli (2006).

ACKNOWLEDGEMENTS

Research partially funded by the grants NSF IIS #1817046 and DTRA #HDTRA1-19-1-0017.

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

| Symbol | Meaning |
|:---:|:---:|
| $\mathcal{G}$ | Flow graph |
| $\mathcal{V}$ | Set of vertices in $\mathcal{G}$ |
| $n$ | Size of $\mathcal{V}$ |
| $\mathcal{E}$ | Set of edges in $\mathcal{G}$ |
| $m$ | Size of $\mathcal{E}$ |
| $\mathcal{E}' \subseteq \mathcal{E}$ | Set of observed edges |
| $m'$ | Size of $\mathcal{E}'$ |
| $\mathcal{X} \in \mathbb{R}^{m \times d}$ | Edge feature matrix |
| $\mathcal{X}[e] \in \mathbb{R}^d$ | Features of edge $e$ |
| $\mathbf{f} \in \mathbb{R}^m$ | Complete flow vector |
| $f_e \in \mathbb{R}$ | Flow for edge $e$ |
| $\hat{\mathbf{f}} \in \mathbb{R}^{m'}$ | Observed flow vector |
| $\hat{f}_e \in \mathbb{R}$ | Observed flow for edge $e$ |
| $B$ | Incidence matrix of $\mathcal{G}$ |
| $\Phi(\mathbf{f}, \mathcal{X}; \mathbf{f}^{(0)}; \Theta) \in \mathbb{R}_+$ | Regularization function |
| $\mathbf{f}^{(0)} \in \mathbb{R}^m$ | Flow prior |
| $\Theta$ | Regularization parameters |
| $\Omega$ | Domain of $\mathbf{F}$ |
| $\mathbf{x} \in \mathbb{R}^{m-m'}$ | Estimated vector of missing flows |
| $\hat{\mathbf{x}} \in \mathbb{R}^{m-m'}$ | True vector of missing flows |
| $\mathbf{x}^{(0)} \in \mathbb{R}^{m-m'}$ | Prior for missing flows |
| $H \in \mathbb{R}^{m \times m'}$ | Map from $\mathbf{f}$ to $\mathbf{x}$ |
| $\widetilde{\mathbf{f}} \in \mathbb{R}^m$ | Vector with observed flows or 0 otherwise |
| $\mathcal{Q}(\mathcal{X}; \Theta) \in \mathbb{R}^{(m-m') \times (m-m')}$ | Regularization function (diagonal matrix) |
| $K$ | Number of folds in cross-validation |
| $T$ | Number of outer iterations for Algorithm 1 |
| $J$ | Number of inner iterations for Algorithm 1 |
| $\alpha$ | Outer learning rate for Algorithm 1 |
| $\beta$ | Inner learning rate for Algorithm 1 |
| $\Psi(\mathbf{x}, \Theta)$ | Outer objective |
| $\Upsilon_\Theta(\mathbf{x})$ | Inner objective |
| $\Gamma_j(\mathbf{x}_{k,j-1}, \Theta_{i-1})$ | One step of SGD |
| $\overleftarrow{\Theta}$ | Estimate of $\nabla_\Theta \Psi(\mathbf{x}, \Theta_{i-1})$ |
| $H_k$ | Matrix $H$ for fold $k$ |
| $Q_k$ | Matrix $Q$ for fold $k$ |
| $\widetilde{\mathbf{f}}_k$ | Vector $\widetilde{\mathbf{f}}$ for fold $k$ |

Table 2: Table of the main symbols used in this paper.

## A  TABLE OF SYMBOLS

Table 2 lists the main symbols used in our paper.

## B  BILEVEL OPTIMIZATION WITH GRAPH NEURAL NETWORKS

This section is an extension of Section 3.3. Here, we consider the case where $\mathcal{Q}(\mathcal{X}; \Theta)$ is a GNN:

$$\mathcal{Q}(\mathcal{X}, \Theta) = diag(GNN(X, \Theta, \mathcal{G})) \tag{8}$$

For instance, we apply a 2-layer spectral Graph Convolutional Network (GCN) with Chebyshev convolutions (Defferrard et al., 2016; Kipf & Welling, 2016; Hammond et al., 2011):

$$\mathcal{Q}(\mathcal{X}; \Theta) = diag\left( ReLU\left( \sum_{z'=1}^{Z'} T_{z'}(\widetilde{L}) ReLU\left( \sum_{z=1}^{Z} T_z(\widetilde{L}) \mathcal{X} W_z^{(1)} \right) W_{z'}^{(2)} \right) \right) \tag{9}$$

where $\widetilde{L} = 2/\lambda_{max}L - I$, $L$ is the normalized Laplacian of the undirected version of the line graph $\mathcal{G}'$ of $\mathcal{G}$, $\lambda_{max}$ is the largest eigenvalue of $L$, $T_z(\widetilde{L})$ is a Chebyshev polynomial of $\widetilde{L}$ with order $z$ and $W_z^{(i)}$ is the matrix of learnable weights for the $z$-th order polynomial at the layer $i$. In a line graph, each vertex represents an edge of the undirected version of $\mathcal{G}$ and two vertices are connected if their corresponding edges in $\mathcal{G}$ are adjacent. Morever $L = I - D^{-1/2}AD^{-1/2}$, where $A$ and $D$ are the adjacency and degree matrices of $\mathcal{G}'$. Chebyshev polynomials are defined recursively, with $T_z(y) = 2yT_{z-1}(y) - T_{z-2}(y)$ and $T_1(y) = y$.

In our experiments, we compare GCN against MLP regularization functions. We have also applied the more popular non-spectral graph convolutional operator (Kipf & Welling, 2016) but preliminary results have shown that the Chebyshev operator achieves better performance in flow estimation.

## C  EXTENDED EXPERIMENTAL SECTION

This section in an extension of Section 4.

### C.1  MORE DETAILS ON DATASETS

**Traffic:** Flow data was collected from the Caltrans—the California Department of Transportation—PeMS (Performance Measurement System).[1] Sensors are placed at major highways in the state. We use sensor geo-locations and other attributes to approximately match them to a compressed version of road network extracted from Openstreetmap.[2] The compression merges any sequence of segments without a branch, as these extra edges would not affect the flow estimation results. We emphasize that this dataset is not of as high quality as Power, due to possible sensor malfunction and matchings of sensors to the wrong road segments. This explains why flow estimation is more challenging in Traffic. Figure 4 is a visualization of our traffic dataset with geographic (lat-long) located vertices and colors indicating light versus heavy traffic (compared to the average). The road segments in the graph (approximately) cover the LA County area. We show the map (from Openstreetmap) of the area covered by our road network in Figure 5.

**Power:** We will provide more details on how we build the power dataset. PyPSA (Python for Power System Analsys) is a toolbox for the simulation of power systems (Brown et al., 2017). We applied the European transmission system (PyPSA-Eur), which covers the ENTSO-E area (Hörsch et al., 2018), to generate a single network snapshot. Besides the PyPSA-Eur original set of edges, which we will refer to as line edges, we have added a set of bus edges. These extra edges allow us to represent power generation and consumption as edge flows. For the line edges, we cover the following PyPSA attributes (with their respective PyPSA identifiers[3]): reactance (x), resistance(r), capacity (s_nom), whether the capacity s_nom can be extended (s_nom_extendable), the capital cost of extending s_nom (capital_cost), the length of the line (length), the number of parallel lines (num_parallel) and the optimized capacity (s_nom_opt). For bus lines, the only attribute is the control strategy (PQ, PV, or Slack). Notice that we create a single vector representation for both line and bus lines by adding an extra indicator position (line or bus). Moreover, categorical attributes (e.g., the control strategy) were represented using one-hot encoding. Figure 6 is a visualization of our power dataset with geographic (lat-long) located vertices and colors indicating high versus low power (compared to the average).

### C.2  EVALUATION METRICS

We apply the following evaluation metrics for flow estimation. Let $\mathbf{f}_{true}$ and $\mathbf{f}_{pred}$ be $m'$-dimensional vectors with true and predicted values for missing flows associated to edges in $\mathcal{E} \setminus \mathcal{E}'$.

*Correlation (Corr):*
$$cov(\mathbf{f}_{pred}, \mathbf{f}_{true})/(\sigma(\mathbf{f}_{pred}).\sigma(\mathbf{f}_{true}))$$

where $cov$ is the covariance and $\sigma$ is the standard deviation.

---

[1]Source: http://pems.dot.ca.gov/

[2]Source: https://www.openstreetmap.org

[3]https://pypsa.readthedocs.io/en/latest/components.html

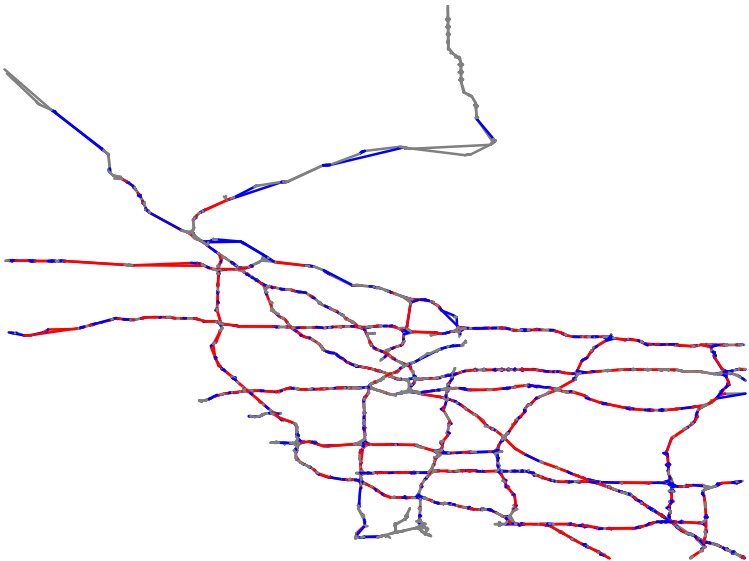

Figure 4: Visualization of our traffic network with geo-located vertices. Edges in grey have missing flows, edges in red have traffic above the average and edges in blue have traffic below the average. Better seen in color. See Figure 5 for map of the area.

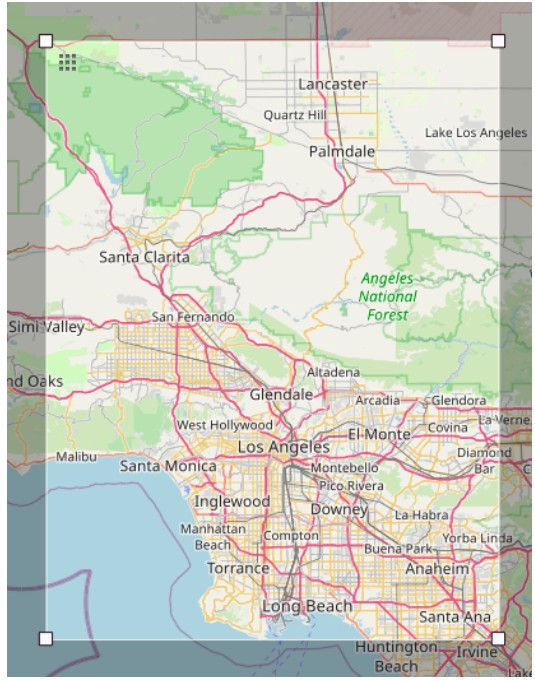

Figure 5: Road map covered by the road network shown in Figure 4 (from Openstreetmap)

*Mean Absolute Percentage Error (MAPE):*

$$\frac{1}{m - m'} \sum_{e \in \mathcal{E} \setminus \mathcal{E}'} |\frac{(\mathbf{f}_{true})_e - (\mathbf{f}_{pred})_e}{(\mathbf{f}_{true})_e}|$$

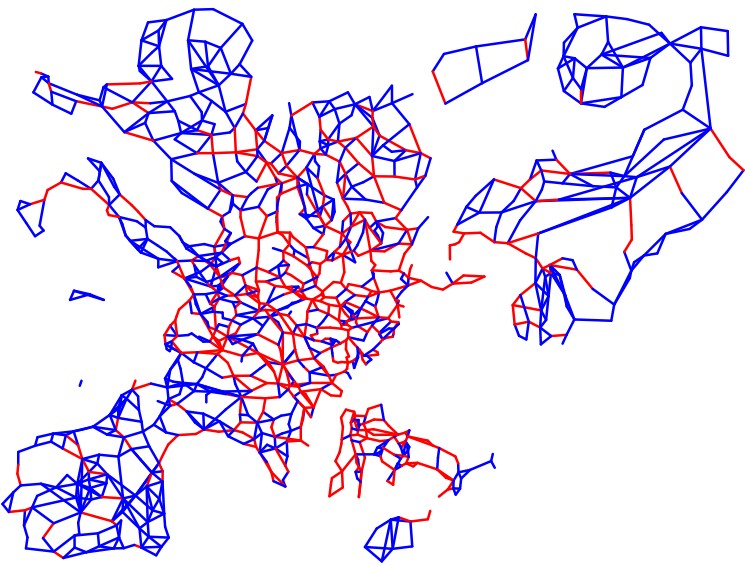

Figure 6: Visualization of our power network with geo-located vertices. Edges in red have traffic above the average and edges in blue have traffic below the average. Better seen in color.

*Mean Absolute Error (MAE):*

$$\frac{1}{m'} \sum_{e \in \mathcal{E} \setminus \mathcal{E}'} |(\mathbf{f}_{true})_e - (\mathbf{f}_{pred})_e|$$

*Root Mean Squared Error (RMSE):*

$$\sqrt{\frac{1}{m'} \sum_{e \in \mathcal{E} \setminus \mathcal{E}'} [(\mathbf{f}_{true})_e - (\mathbf{f}_{pred})_e]^2}$$

*Divergence (Div):*

$$\sum_v (\sum_u \mathbf{f}_{(u,v)} - \sum_u \mathbf{f}_{(v,u)})^2$$

## C.3 MORE EXPERIMENTAL SETTINGS

**Train/test splits:** We report results of a 10-fold cross-validation based on the set of labeled flows. Moreover, we use 10% of training flows for validation.

**Implementation[4]:** We have implemented Algorithm 1 using PyTorch, CUDA, and Higher (Grefenstette et al., 2019), a meta-learning framework that greatly facilitates the implementation of bilevel optimization algorithms by implicitly performing the reverse iterations for a list of optimization algorithms, including SGD. Moreover, our GCN implementation is based on the Deep Graph Library (DGL) (Wang et al., 2019).

**Hardware:** We ran our experiments on a single machine with 4 NVIDIA GeForce RTX 2080 GPUs (each with 8Gb of RAM) and 32 Intel Xeon CPUs (2.10GHz and 128Gb of RAM).

---

[4]https://github.com/arleilps/flow-estimation

**Hyperparameter settings:** We have selected the parameters based on RMSE for each method using grid search with learning rate over $[10^0, 10^{-1}, 10^{-2}, 10^{-3}]$ and number of nodes in the hidden layer over $[4, 8, 16]$. The total number of iterations was set to 3000 for Min-Div and 5000 for MLP and GCN, all with early stop on convergence after 10 iterations. For our methods (both based on Algorithm 1), we set $T = 10, J = 300, \alpha = 10^{-2}, \beta = 10^{-2}$ and $K = 10$ in all experiments.

### C.4 DIVERGENCE RESULTS

Although the main goal of flow estimation is to minimize the flow prediction loss, we also evaluate how our methods and the baselines perform in terms of divergence (or flow conservation) in Table 3. As expected, MLP and GCN do not conserve the flows. However, interestingly, our methods (Bil-MLP and Bil-GCN) achieve higher flow conservation than Min-Div. This is due to the regularization parameter $\lambda$, which is tuned based on a set of validation flows.

|  | Traffic | Power |
|---|---|---|
| Min-Div | 2.94 | 2.45 |
| MLP | 5.69 | 2.77 |
| GCN | 5.71 | 2.80 |
| Bil-MLP | 2.81 | 2.43 |
| Bil-GCN | 2.83 | 2.43 |
| Bil-GCN-Prior | 2.43 | 2.43 |

Table 3: Divergence results.

## D TRUE VS. PREDICTED FLOWS

Figure 7 shows scatter plots with the true vs. predicted flows that are missing from Figure 2.

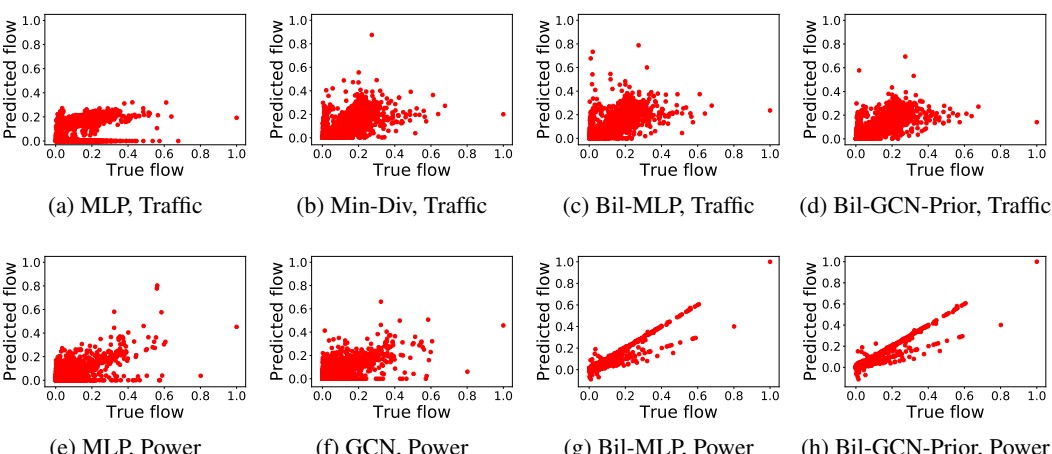

Figure 7: Scatter plots with true (x) and predicted (y) flows for remaining methods (beyond the ones shown in Figure 2).

### D.1 VISUALIZATION OF REGULARIZER FOR POWER

Figure 8 shows the regularizers over the Power network topology. As discussed in Section 4.4, the regularizer affects mostly a few top resistance edges. For the remaining ones, regularizers have a small value. Notice that these high resistance edges are associated with lines transmitting small amounts of power, as shown in Figure Figure 6, and have a large impact on the overall flow estimation accuracy.

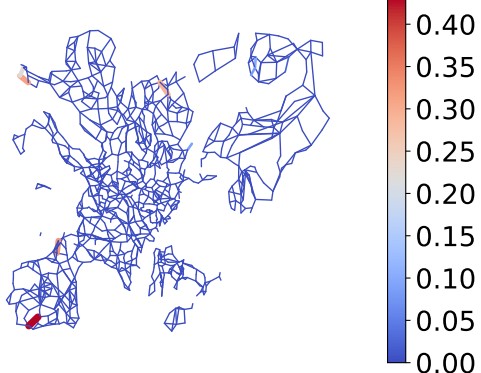

Figure 8: Visualization of regularizers on the Power network topology. We highlight edges with large vaues of regularizer. Better seen in color.

| Method | Traffic | | Power | |
|---|---|---|---|---|
| | **Training** | **Test** | **Training** | **Test** |
| Min-Div | 424.4 | 0.09 | 364.2 | 0.01 |
| MLP | 21.95 | 0.10 | 12.32 | 0.01 |
| GCN | 2.43 | 0.09 | 0.77 | 0.01 |
| Bil-MLP | 1860.2 | 0.08 | 369.7 | 0.01 |
| Bil-GCN | 1870.1 | 0.09 | 346.7 | 0.01 |
| Bil-MLP-Prior | 1886.1 | 0.01 | 334.1 | 0.01 |

Table 4: Average training and test times (in seconds) for our methods and the baselines (in seconds).

## D.2 RUNNING TIME

Table 4 shows the average running times—over the 10-fold cross-validation—of our methods and the baselines for the Traffic and Power datasets. We show both training and test times. The results show that our reverse-mode differentiation algorithm adds significant overhead on training time for Traffic, taking up to 4 times longer than Min-Div to finish. As described in Section 3.4, this is due mainly to the cost of computing and storing the inner problem iterations. On the other hand, all the methods are efficient at testing. GCN converged quickly (due to early stopping) for both datasets. However, it achieved poor results for Power, as shown in Table 1, which is a sign of overfitting or underfitting. Notice that the results reported are the best in terms of RMSE.

