# OpenReview forum: "Combining Physics and Machine Learning for Network Flow Estimation"
_ICLR.cc/2021/Conference — ICLR 2021 Poster_

### Official Review · AnonReviewer1 · 2020-10-26
**A decent illustration of PIML in the vein of UDE.**

**Rating:** 7
**Confidence:** 3

**Review:**

The authors propose a parametric regularizer for estimating unobserved flows in networks, incorporating edge features and other side information. The parameters of the regularizer are learned by means of minimizing the empirical cross-validated MSE. Regularization is necessary because the basic problem, while convex, typically is under-constrained; resulting in a infinite space of solutions which match the observed data.

The paper is clearly written and easy to follow. It conceptually follows the universal-differential-equation (UDE) framework for integrating physics and machine learning, in which certain elements of a physical model are replaced by a parametric model which can be trained from data. This connection makes the work topical, and a small step forward in physics-informed machine learning.

For this particular problem, it seems that (i) the prior is indispensable, some prior (perhaps implicit) will be contained in any regularization, and (ii) the learned parameters may depend implicitly on the prior. This brings up two difficulties. Where does one obtain a (good) prior if one does not have one already? And, how can we hope the parameterized regularizer will generalize for new priors? Perhaps by virtue of it's simplicity?

Other potential issues I should like to have seen addressed are:

The model is well suited for instantaneous flows. For the traffic problem, it seems it is predicting current flow from current flow. Yet, it takes time for cars to move about. It would be nice to see temporal dynamics included.

Only diagonal regularizers are considered. Can the authors comment on learning more general regularizing matrices?

Can the proposed learning algorithm (or similar) be used to inform which edges ought to be measured to maximize one's ability to impute the missing flows? If so, how? If not, why?

Finally, and not least, the cross-validation routine appears to equally weight each edge. Yet, some edges may be more constrained than others. Certainly, each fold will have it's own, potentially different, infinite subspace of solutions to the unregularized problem. Failing to account for this may bias the learning algorithm. One may see undue performance degradation when applying a trained model to a different pattern of observed flows for the same network.

---

> ### Author Response · Authors · 2020-11-25
> **Priors, generalizations, active learning, bias**
>
> Thanks for the comments!
>
> Two potential priors for our model would be existing machine learning models that disregard the physics and previous flow observations on a particular edge (this is the case for most traffic inference models, such  as [1]). Regarding changing priors, we do expect that different flow distributions might require different regularizers. For instance, in the case of traffic, if the data contains only low-usage times (free flow), the model might be unable to learn the effect of the number of lanes on traffic at an edge. Thus, we do assume that the training data is, to a certain  extent, representative of the true flows.  We have added a sentence about this when we define priors.
>
> [1] Li et al. Diffusion Convolutional Recurrent Neural Network: Data-Driven Traffic Forecasting. ICLR’18
>
> We have added generalizing our model to the dynamic setting as future work.
>
> In principle, one could extend our framework to the case of a full regularization matrix. Using the Bayesian interpretation, the off-diagonal terms are related to off-diagonal entries of a covariance matrix. However, the physical interpretation of such terms, which one would expect to depend on features of pairs of edges, is not clear. One possibility is for them to capture correlated flow values across the network. A more practical challenge is how to deal with the much larger number of parameters to be learned. One could assume such a matrix to be sparse or to consider other priors to avoid overfitting. We have added the case of full regularization matrices as future work.
>
> One possible heuristic for active learning is to use a validation set of flows to compute flow residuals (in absolute values) for the corresponding edges and then estimate the residuals for missing edges by solving the flow estimation problem over the residuals. We have not been able to implement this approach yet, but, in case it is effective, we will add the results to the final version of the paper. We also want to compare this approach against alternatives based only on the graph topology, such as those proposed in [1].
>
> [1] Jia et al. Graph-Based Semi-Supervised and Active Learning for Edge Flows. KDD. 2019.
>
> We agree with the reviewer that learning regularizers that generalize to multiple snapshots might be a challenge. One way to address this problem is to learn regularizers jointly over a large enough sample of snapshots. Notice that this task can be solved using our model by treating each snapshot as a disconnected component. Because the same edge in two snapshots will share the same attributes, they must also share their value of regularizer. We plan to add some experiments illustrating this to our final version.

---

### Official Review · AnonReviewer2 · 2020-10-27
**Interesting topic with limited novelty**

**Rating:** 4
**Confidence:** 5

**Review:**

In this paper, the authors introduce a method for missing flow estimation. These method has potential to address some important applications in transportation, power systems and water management. One major difference compared with the previous work is that edge features are incorporated into the optimization process so that the model has a better chance at learning edge-specific patterns. The experimental results have shown some success of the proposed method in traffic and power datasets.

There are several major issues with the paper:
1. The model is based on the assumption of flow conservation. However, this is not always true in real-world scenarios. The power can be generated or consumed while water flow can also be affected by precipitation.

2. The novelty is limited as most parts are based on existing work. The authors claim that this method combines physics and ML but I did not see physics being used anywhere in designing the model (except for the conservation). The use of matrix Q (and edge features) may not necessarily learn patterns that are consistent to underlying physics (since it is still learned automatically).

3. Notation used in the paper is confusing, even if people can still get most of it after reading through the entire paper. For example, authors may want to include the definition of x (missing flows) in Section 2. In Eq. 1, do you need the operator \sum_vi?

4. An alternative way to think about this problem is to train a predictive model that predicts x from edge features (maybe using a graph neural net) and then enforce the flow conservation. What is the difference between this predictive framework and the proposed bilevel optimization framework? It would be great to compare with other predictive method.

---

> ### Author Response · Authors · 2020-11-25
> **Assumptions, novelty, physics+ML, notation, baselines**
>
> Thanks for the comments!
>
> Known inputs and outputs can be easily accounted for in the model as observed flows. In case inputs and outputs are only approximately known, they can be given as priors in our model. Moreover, in the more realistic case that many inputs and outputs are unknown, flow conservation might be our best approximation of the system and our model will recover that. We have added a discussion on our flow conservation assumption to Section 2.
>
> Our work is part of a recent effort to incorporate physics knowledge into machine learning models (see next answer). Our model is built upon existing work (bilevel optimization, inverse problems, regularized least squares, standard and graph neural networks, among others). However, the combination of these different techniques to solve the flow estimation problems is novel given that existing machine learning models for flow graphs are unable to account for flow conservation and learn to regularize from data. We claim that this is a significant contribution given the broad applications of flow graphs (road, power, water etc.) in the sciences and engineering. We have improved our introduction to highlight the novelty of our work.
>
> [1] Bressan et al. Flows on networks: recent results and perspectives. EMS. 2014.
>
> [2] Garavello and Piccoli. Traffic Flow on Networks: Conservation Laws Model. AMS. 2006.
>
> [3] Li et al. Diffusion Convolutional Recurrent Neural Network: Data-Driven Traffic Forecasting. ICLR. 2018.
>
> [4] Jia et al. Graph-Based Semi-Supervised and Active Learning for Edge Flows. KDD. 2019.
>
> The key assumption of our model is that the physics of flow networks depends on a conservation law and edge features. First, the role played by conservation laws (energy, mass, momentum, charge etc.) in the sciences in general should not be underestimated (see Noether’s theorem, for instance). On the other hand, existing ML models for inference on physical networks, especially for traffic, capture only correlations between speed values, disregarding flows and their conservation. Because flow conservation alone makes the flow estimation problem ill-posed (it is an inverse problem)  regularization is key to enable the recovery of the model parameters based on domain-knowledge. By incorporating domain-specific physics to machine learning, we can increase its impact on fields where data availability is limited, as is often the case in engineering and the natural sciences. We have improved our motivation describing how our model is able to combine physics and ML in our introduction.
>
> [1] Hanca et al. Symmetries and conservation laws: Consequences of Noether's theorem. AJP. 2004
>
> [2] Arridge. Solving inverse problems using data-driven models. Acta Numerica. 2019.
>
> Augmenting physical models with machine learning has become a popular approach to bridge the gap between physics and data in many applications. In our model, the matrix Q regularizes the solution of the flow estimation problem. This is important because flow conservation alone makes flow estimation ill-posed. Regularization is known to enable the expression of domain-knowledge in the solution of ill-posed problems and our model learns this regularization function from data. Our results also show that the learned regularization function is able to learn patterns that are consistent with the physics of flow networks.  We have improved our motivation describing how our model is able to combine physics and ML in our introduction.
>
> [1] Rackauckas et al. Universal differential equations for scientific machine learning. Arxiv. 2020.
>
> [2] Engl et al. Regularization of inverse problems. Springer. 1996.
>
> [3] Arridge. Solving inverse problems using data-driven models. Acta Numerica. 2019.
>
> We reviewed the notation of the paper carefully and addressed the comments raised by the reviewer and many others.
>
> Different from our model, the one proposed by the reviewer is unable to learn edge regularizers, which capture the effect of edge features over the flows (e.g. how numbers of lanes and speed limits constraint traffic). We claim that such regularizers are part of the physics of flow networks and a key contribution of our paper is the ability to learn them from data. Instead, the model proposed by the reviewer can only learn correlations between edge features and flows (e.g. road segments with more lanes have higher traffic). To illustrate the difference between these two approaches we have added baselines based on the idea given by the reviewer. More specifically, they combine the approach by Jia et al. with priors learned using an MLP and GCN. The weight given by the MLP/GCN prediction versus flow conservation is optimized using validation data (see Section 4.2 and Table 1 for details). These approaches achieve similar performance to the best between divergence minimization and MLP/GCN but are not significantly better.
>
> [1] Jia et al. Graph-Based Semi-Supervised and Active Learning for Edge Flows. KDD. 2019.

---

### Official Review · AnonReviewer3 · 2020-10-28
**somewhat not fitting the conference title**

**Rating:** 6
**Confidence:** 5

**Review:**

authors study the problem of learning flows form partially observed flow networks. The estimationg of unobserved flows is based on flow conservation and edge features.

only little theorerical analysis is offered:
- does the proposed method converge?
- is the learning problem (bi-level optimization problem) intriniscally hard (non-convex)?

im also missing a discussion of how the proposed method relates to min-cost flow problems which are quite well-studiedl see the textbook Network Optimization: Continuous and Discrete Models  by Bertsekas.
A recent line of work connects min-cost flow problems with non-smooth convex optimization:

A. Jung, "On the Duality Between Network Flows and Network Lasso," in IEEE Signal Processing Letters, vol. 27, pp. 940-944, 2020, doi: 10.1109/LSP.2020.2998400.

the numerical experiments are not very convincing. Also, it is unclear how the edge features are obtained precisely for the traffice network example. It is mentioned that we know in-degree, out-degree and number of lanes. How is the in-degree of an edge defined?

---

> ### Author Response · Authors · 2020-11-25
> **Fitting, convergence, related problems**
>
> Thanks for the comments!
>
> Our paper is related to several topics that have historically attracted interest from the ICLR community, such as applications of deep learning to traffic data, bilevel optimization, meta-learning and physics-informed neural networks:
>
> [1] Li et al. Diffusion Convolutional Recurrent Neural Network: Data-Driven Traffic Forecasting. ICLR’18
>
> [2] Mackay et al. Self-Tuning Networks: Bilevel Optimization of Hyperparameters using Structured Best-Response Functions. ICLR’19
>
> [3] Yin et al. Meta-learning without memorization. ICLR’20
>
> [4] Seo et al. Physics-aware Difference Graph Networks for Sparsely-Observed Dynamics. ICLR’20
>
> Bilevel optimization is non-convex and thus we cannot guarantee that Algorithm 1 will return a global optima. In particular, the learning objective of our regularization function is non-convex---it is a neural network. However, the inner problem (Equation 5) in our formulation has a convex objective (least-squares). A convergence analysis for RMD [3] has shown that this property implies convergence. We also find that our algorithm often converges to a good estimate of the parameters in our experiments. We have added some discussion on convexity and convergence of our algorithm to Sections 3.3 and 3.4.
>
> [1] Colson et al. An overview of bilevel optimization. Annals of Operations Research. 2007.
>
> [2] Franceschi et al. Forward and Reverse Gradient-Based Hyperparameter Optimization. ICML. 2017.
>
> [3] Franceschi et al. Bilevel Programming for Hyperparameter Optimization and Meta-Learning. ICML. 2018.
>
> There are a few key differences between our formulation and classical flow optimization problems, such as min-cost flows and multi-commodity flows. First, we don’t assume known costs and capacities for edges in the graph and instead learn edge regularization functions from data. Second, we don’t enforce flow conservation as a hard constraint, but add it to the objective (as a soft constraint). Our approach is more flexible to noise in observed flows and unknown source/sinks in the graph. We have added a discussion on classical flow optimization problems to the related work.
>
> Jung's work shows the duality between node-level inference (for graph signals) and min-cost (edge) flows. As discussed earlier, there are key differences between classical flow optimization and our model and thus the same reasoning cannot be directly applied to devise a duality relation for flow estimation. Moreover, in his formulation flows capture node value discrepancies on edges (e.g. equal value means no flow). In the general case, such as when some flows are given as inputs, flows cannot be directly mapped to smooth node functions. We have added a discussion on classical flow optimization problems and node-level inference to the related work.
>
> Given that the reviewer has not offered any specific comment regarding the experiments, we have conducted a detailed checking of the data, experiments and results. We believe , to the best of our capacity, that all the results are correct and will also share the code and data used in our experiments.
>
> The in-degree of a directed edge is defined as the in-degree of its source node and the out-degree is defined as the out-degree of its destination node. We have added these definitions to the paper.

---

### Official Review · AnonReviewer4 · 2020-10-31
**Interesting idea of incorporating the flow-preservation with promising results.**

**Rating:** 7
**Confidence:** 3

**Review:**

This paper investigates the very important problem of incorporating physics into machine learning model with special focus on the missing flow estimation problem. It proposes the a flow-estimation model that combines the reverse-model differentiation and neural network considering the edge features. Empirical results on two real-world datasets show very promising results with interpretable physical properties.

Clean presentation and easy to follow.

Strength:
- The investigated problem of incorporating physics into machine learning model is very important.
- The idea is combining reverse-mode differentiation into the modeling is intuitive and also shown to be effective.
- The proposed algorithm achieves improved results and also show intuitive interpretation.
- The paper provides detailed information implementation about experiments, which make the result more convincing and easier to reproduce.

Here are some minor concerns:
- The proposed algorithm applies to problems on the flow graphs. Yet, for many applications on flow graphs, e.g., traffic speed (instead of flow) prediction, it may not be straight-forward to apply the proposed algorithm.
- It is will be interesting to see if the flow preservation can be incorporated directly in the model forward path of training instead of using a regularizer.

---

> ### Author Response · Authors · 2020-11-25
> **Flow conservation and objective**
>
> Thanks for the comments!
>
> We agree with the reviewer that there are edge values, such as speed, that are not governed by a conservation law. However, physical models for traffic (see references below) relate speed, flow and density under the assumption of flow conservation via the so-called fundamental diagram. Thus, according to these physical models, machine learning models that predict only speed values cannot fully capture the physics of traffic. In the future, we want to extend our model to account for multiple edge/node measurements, including speeds. We have added more details on this topic as a future work in our conclusion.
>
> [1] Lighthill and Whitham.  On kinematic waves II. A theory of traffic flow on long crowded roads. PRSL. 1955.
>
> [2] Richards. Shock waves on the highway. Operations research. 1956.
>
> [3]  Garavello and Piccoli. Traffic Flow on Networks: Conservation Laws Model. AMS. 2006.
>
> [4] https://en.wikipedia.org/wiki/Fundamental_diagram_of_traffic_flow
>
> While imposing strong flow conservation constraints might be useful for flow estimation and there are alternatives to regularization for constrained deep learning, we find that having it as part of the object is effective due to noise in observed data and possibly unknown inputs/outputs in the flow graph.  In particular, the problem might not have a solution under flow conservation constraints. On the other hand, we have optimized the parameter lambda of our Min-Div baseline, which covers the case of flows approximately conserved (when lambda is small). We have added a discussion on soft vs hard constraints in Section 2.

---

### Decision · Program_Chairs · 2021-01-07
**Final Decision**

**Decision:**

Accept (Poster)

**Comment:**

The paper combines bi-level optimization and reverse mode-differentiation for flow estimation on networks. Most reviewers think the idea of incorporating physical constraints is interesting and novel, and the experiments convincing.